# Invasive Streptococcal Infection in Children: An Italian Case Series

**DOI:** 10.3390/children11060614

**Published:** 2024-05-21

**Authors:** Francesca Rivano, Martina Votto, Silvia Caimmi, Patrizia Cambieri, Riccardo Castagnoli, Marta Corbella, Mara De Amici, Maria De Filippo, Enrico Landi, Antonio Piralla, Ivan Taietti, Fausto Baldanti, Amelia Licari, Gian Luigi Marseglia

**Affiliations:** 1Department of Clinical, Surgical, Diagnostic and Pediatric Sciences, University of Pavia, 27100 Pavia, Italyenrico.landi01@universitadipavia.it (E.L.); gianluigi.marseglia@unipv.it (G.L.M.); 2Pediatric Clinic, Fondazione IRCCS Policlinico San Matteo, 27100 Pavia, Italy; 3Microbiology and Virology Department, Fondazione IRCCS Policlinico San Matteo, 27100 Pavia, Italya.piralla@smatteo.pv.it (A.P.); 4Laboratory of Immuno-Allergology of Clinical Chemistry, Fondazione IRCCS Policlinico San Matteo, 27100 Pavia, Italy

**Keywords:** group A streptococcus, pharyngitis, invasive infection, children, scarlet fever

## Abstract

Since October 2022, alerts have spread from several countries about the increase in invasive group A streptococcal (iGAS) and scarlet fever cases affecting young children. We aim to analyze the epidemiology of GAS infections in the last 12 years in our hospital and identify the clinical features of invasive cases observed in 2023. We conducted a retrospective study enrolling children and adolescents hospitalized at our pediatric clinic from January to December 2023 for a definitive diagnosis of iGAS infection. Clinical, laboratory, and imaging data were collected and analyzed. Comparing 2016 and 2023, we observed a similar number of GAS infections (65 vs. 60 cases). Five children with iGAS infection were hospitalized between March and April 2023. The median age was five years. At admission, all patients showed tachycardia disproportionate to their body temperature. Vomiting was a recurrent symptom (80%). Laboratory tests mostly showed lymphopenia, hyponatremia, and high inflammatory markers. The number of pediatric iGAS cases significantly increased in 2023. Clinical (pre-school-aged children with high fever, unexplained tachycardia, and vomiting) and laboratory parameters (high procalcitonin levels, hyponatremia, and lymphopenia) could help identify and suspect a potential iGAS infection.

## 1. Introduction

Historically, invasive group A streptococcal (iGAS) infections have always been a relevant cause of morbidity and mortality [1,2,3]. Since October 2022, several alerts have spread from European countries, the US, and Australia concerning the increase in GAS and scarlet fever cases in children [4,5,6,7,8,9,10,11,12,13,14,15,16,17].

Notifications of scarlet fever, GAS pharyngitis, and iGAS cases from the United Kingdom Health Security Agency (UKHSA) have significantly increased from September to early December 2022 compared to the previous years [4,6,7]. The increase in iGAS infections was observed among children, representing about a quarter of the total cases (26.1%, compared to 6.4–13.3% in previous years). About 63% of affected children were males. Proportionally, deaths due to iGAS infections increased. GAS was mostly isolated from pleural fluid and the lower respiratory tract among reported cases. Interestingly, this increase was not seen in older patients [4,7]. In Scotland, a rise in pleural empyema cases associated with GAS infection was reported from September to December 2022, with high rates of cases requiring chest drainage. The patient ages ranged from 10 months to 11 years; 56% were boys, and only two children had pre-existing medical conditions, including prematurity. Nine of these patients required intensive care, but there were no deaths [6]. Reports from England and Wales described a more frequent association between iGAS and respiratory viral infections, although the prevalence of viral diseases was stable. The most frequent coinfections were Respiratory Syncytial virus (RSV), human Metapneumovirus (hMPV), and Rhinovirus [4,6,7].

In the Netherlands, de Gier et al. described a significant surge in iGAS cases from March 2022. They notified 42 pediatric cases (70% 0–5 years) of iGAS in 2022, a significantly higher number than in the previous years (six cases in 2016–2019, three in 2020, and two in 2021); there were nine fatal cases. Seven patients had a previous or concurrent Varicella Zoster virus (VZV) infection [8]. Van Kempen et al. conducted a survey involving seven hospitals to assess the epidemiology of pediatric iGAS cases admitted between July 2021 and June 2022, compared to the pre-pandemic phase (January 2018–December 2019). They found the most significant increase occurred in children younger than five years. Moreover, the clinical presentation of iGAS changed throughout the study period. Pneumonia with empyema was prevalent in 2021–2022 (28%). Instead, sepsis was more common in 2018–2019 (21%). Mortality was also higher in 2021–2022 (9% vs. 3%) than in 2018–2019. A previous VZV infection was documented in 16 cases, whereas Influenza virus infection was reported in 9 cases [9]. 

In Spain, Cobo-Vázquez et al. [10] analyzed the features of children with iGAS infection in 24 hospitals (PedGAS-net). They found a significant increase in iGAS infections between November and December 2022. The median age of affected children was 41 months, and the most common clinical picture was pneumonia, often associated with RSV and flu infections [10,11]. 

In France, 121 cases of iGAS infection were admitted to the PICU between September 2022 and January 2023. Most children were male, aged 1–4 years. Eleven children died. About two-thirds of the cases presented with lower respiratory tract infections, and almost 55% of patients presented with a viral coinfection, most often Influenza virus (25%) and RSV (19%) infection [12].

Barnes et al. analyzed data from the CDC’s Emerging Infections Program in Colorado and Minnesota. They compared the pre-pandemic (January 2016–December 2019), the pandemic (January 2020–December 2021), and the recent period (October 2022-December 2022). Thirty-four iGAS cases were reported in October–December 2022, compared to 11 and 4 cases in the pre-pandemic and pandemic phases. The median age of children was three years in Colorado and 6.5 in Minnesota. PICU admission was similar to the previous years (35% vs. 34%). Twenty-one patients had a viral coinfection, most frequently RSV, Influenza virus, and Severe Acute Respiratory Syndrome-Coronavirus (SARS-CoV)—2 [13,14].

In Australia, Abo et al. reported data from The Paediatric Active Enhanced Disease Surveillance (PAEDS). This network notified a significant rise in iGAS cases (103 cases) from the first four months of 2022. The median age of children was 4.5 years. Pneumonia and bacteremia were the most frequent clinical manifestations. The proportion of severe cases remained stable through the years. The most commonly identified viruses were Rhinovirus/Enterovirus, hMPV, and Adenovirus [15].

Since January 2023, there has been an increase in scarlet fever cases and streptococcal pharyngitis in Italy [17]. Garancini et al. published the first Italian case series on GAS infections, reporting four cases of iGAS infection requiring PICU admission [18]. In this context, we described the largest Italian pediatric cohort of iGAS infections, referring to a pediatric clinic in Northern Italy. Therefore, this study aims to (i) analyze the epidemiology of GAS infections in the last 12 years in our pediatric clinic and (ii) identify clinical features of invasive cases.

## 2. Materials and Methods

According to the STROBE guidelines [19], we performed a case series of pediatric patients admitted to the Pediatric Clinic of Fondazione IRCCS Policlinico San Matteo in Pavia, Italy, for iGAS infection. Then, we extracted data from electronic medical records concerning all children and adolescents (<16 years) hospitalized for iGAS infection in our Clinic in the last 12 years (January 2012–December 2023). The diagnosis of invasive disease was defined upon identifying GAS in sterile sites.

We collected clinical features, including demographics (age at onset, sex, ethnicity), past medical history (comorbidities, vaccine history), viral and bacterial coinfections, antibiotic therapy, previous emergency department (ED) admission, symptoms (fever, malaise, neurological [headache, drowsiness, lethargy, motor impairment], abdominal [vomiting, diarrhea, abdominal pain], respiratory symptoms [ear pain, sore throat, cough, chest pain, respiratory distress]) and vital parameters (heart rate, body temperature, respiratory rate, blood pressure) at the admission. We also collected laboratory findings at admission (pH, white blood cell count, absolute neutrophil count, absolute lymphocyte count, absolute monocyte count, hemoglobin, platelet count, levels of reactive-C protein [CRP], procalcitonin, alanine and aminotransferases, urea, creatinine, lactate, bicarbonates, sodium) and radiological findings (chest X-ray, brain CT scan, brain vascular CT scan, abdomen US, echocardiography). We also reported therapies. All patients provided written informed consent. The Ethical Committee of Fondazione IRCCS Policlinico San Matteo, Pavia, approved this study. This study is linked to the project INF-ACT (project number PE00000007), supported by EU funding within the NextGeneration EU-MUR PNRR Extended Partnership initiative on Emerging Infectious Diseases. Written informed consent was obtained from all subjects involved in this study.

Qualitative variables were presented as numbers (%), and quantitative variables as the median and interquartile range (IQR). The statistical analyses were performed through Prism 9, GraphPad Statistics software, Version 9.5.1.

## 3. Results

### 3.1. Epidemiology of GAS Infections

In the last 12 years, we reported 431 cases of GAS infection diagnosed with suggestive clinical symptoms (i.e., fever, pharyngitis, scarlet fever) and positive pharyngeal swabs. The highest prevalence of GAS cases was observed in the late winter and spring (Figure 1). The most significant number of GAS infections (n = 65, 15%) was observed in 2016, whereas the second was in 2023 (n = 60, 14%). We reported nine cases of iGAS infection during the study period in the spring and early summer (April–June). Four (44%) occurred before 2023, two (20%) in 2016, one (10%) in 2019, and one (10%) in 2022. Interestingly, the four cases recorded before 2023 were soft tissue infections (Table 1).

The first case (Case 1) involved a 1-year-old patient diagnosed with cellulitis of the right ankle in the course of GAS sepsis. The second case (Case 2) involved a 12-year-old patient admitted for post-traumatic cellulitis of the right knee, with *S. pyogenes* isolation from drainage fluid. In 2019, a 3-year-old patient (Case 3) was admitted for cellulitis of the left lower limb with isolation of *S. pyogenes* from the lesion swab. The case (Case 4) observed in 2022 involved a 10-year-old patient with a post-traumatic abscess under the left knee flap. The knee’s magnetic resonance imaging (MRI) was not associated with osteotendinous and articular district changes. Despite the antibiotic therapy, the patients required surgical drainage, and the synovial liquid culture was positive for *S. pyogenes* infection.

### 3.2. iGAS Infections in 2023

Five (55%) iGAS cases were observed in 2023, constituting 8.3% of *S. pyogenes* infections 2023 (Table 2). Three (60%) patients were males. The median age was five years (IQR 2.5–5.5 years). The peak of cases occurred between March and April 2023. 

A 5-year-old boy (Case 5) was brought to the ED for right otalgia, fever, headache, and vomiting. He was diagnosed with acute otitis media and was discharged in good clinical condition after intravenous (IV) rehydration and antiemetic therapy. The following day, the child was brought to the ED for worsening symptoms. He was sleepy and tachycardic, but blood pressure and respiratory exchanges were normal. Blood tests revealed hyperlacticaemia, hypertransaminasemia, hyponatremia, elevated neutrophil count with lymphopenia, and procalcitonin elevation. Considering a suspected sepsis, he was admitted to the PICU. A brain vascular computer tomography was performed, which demonstrated bilateral mastoiditis complicated by thrombosis of the right sigmoid sinus, jugular bulb, and part of the right internal jugular vein. Brain MRI confirmed mastoiditis with pansinusitis and vascular thrombosis. He subsequently presented a clinical and laboratory picture of muti-organ failure (MOF), characterized by acute liver, heart, renal failure, coagulopathy, and thrombocytopenia, requiring a prolonged pediatric intensive care unit (PICU) treatment. The initial antibiotic treatment empirically included IV ceftriaxone and clarithromycin. The blood culture and drainage secretions from the tympanic membrane were positive for multi-sensitive *S. pyogenes*. This therapy was changed by adding clindamycin to ceftriaxone due to the initial poor improvement in clinical and inflammatory parameters. After about a month, the patient was successfully discharged without permanent sequelae.

A 5-year-old boy (Case 6) was admitted to the ED for fever and vomiting. He was hydrating and feeding adequately. For the finding of expiratory moans, the patient underwent a chest X-ray, which showed bronchitis. He was discharged with azithromycin therapy and oral rehydration. Twelve hours later, the patient returned to the ED for respiratory symptoms and several episodes of vomiting and diarrhea. On physical examination, he presented tachycardia and tachypnea and was hypoxemic, so oxygen therapy was implemented. Blood tests demonstrated elevated neutrophil count with lymphopenia, hyponatremia, and elevation of CRP, especially procalcitonin. A new chest X-ray was performed, which showed left pneumonia with ipsilateral pleural effusion. A broad-spectrum antibiotic treatment with IV ceftriaxone was promptly initiated and continued according to the sensitivity of *S. pyogenes*, found in pharyngeal swab and blood culture. After a short period of respiratory stability with a heated, humidified, high-flow nasal cannula ventilation, the child showed a new worsening of respiratory dynamics due to increased pleural effusion, requiring PICU admission and pleural drainage. *S. pyogenes* was even isolated from the pleural liquid. The antibiotic therapy was optimized with linezolid and metronidazole, with a gradual clinical and radiological improvement.

A 4-year-old boy (Case 7) was admitted to the ED for fever, sore throat, vomiting, and diarrhea. He was born at 35 weeks of gestational age and had a history of multiple congenital malformations (anorectal and esophageal atresia surgically corrected and nonfunctioning left multi-cystic kidney). At the physical examination, he presented a flushed pharynx with hypertrophied tonsils and tachycardia. Blood tests demonstrated hyponatremia and initial hyperlacticaemia, elevated neutrophil count with lymphopenia, elevated CRP, and high procalcitonin levels. The empiric antibiotic therapy with piperacillin-tazobactam was prescribed, suspecting a urinary tract infection. The multi-sensitive *S. pyogenes* was isolated from blood and pharyngeal swab culture. 

A 6-year-old Egyptian girl (Case 8) was admitted to the ED for fever, headache, vomiting, and meningeal signs (nuchal, trunk, and limb stiffness). Blood exams showed hyponatremia, hyperlacticaemia, elevated neutrophil count with lymphopenia, and elevated CRP; procalcitonin was slightly augmented. A brain CT scan was performed, demonstrating pansinusitis. The child was treated with IV ceftriaxone after the lumbar puncture and blood culture. The cerebrospinal fluid culture was positive for *S. pyogenes*. The course of the disease was favorable, and the child completely recovered without complications.

A 1-year-old female child (Case 9) was admitted to the ED of another hospital for fever and respiratory distress. At the clinical evaluation, she presented with compromised general conditions, dyspneic, and hypoxemic. A chest X-ray showed a complete opacification of the right lung due to abundant pleural effusion. Laboratory tests showed marked hyponatremia, neutrophil leukocytosis, and CRP elevation. The PCT value at admission was not performed. Initially supported with low-flow oxygen, IV ampicillin-sulbactam, and steroids, she presented rapid deterioration of respiratory dynamics. For this clinical picture, she was admitted to the PICU of our hospital. *S. pyogenes* was isolated from the pleural culture examination. Considering the initial improvement of symptoms, ampicillin-sulbactam was continued until discharge from the PICU. 

#### Reflections on Reported iGAS Cases 

All (100%) patients presented with fever and tachycardia at the admission. Four (80%) patients had vomiting along with worsening general conditions, and three (60%) presented diarrhea. Laboratory tests showed lymphopenia in four (80%) cases and hyponatremia in all (100%) children. Inflammatory blood markers were high in all (100%) cases. 

The focus of infection was isolated in all patients: pleural fluid in two (40%) children, spinal fluid (20%), middle ear (20%), and pharynx (20%). Bloodstream cultures were positive in three (60%) patients. 

Three (60%) patients required PICU admission. Two (40%) children were admitted to PICU because of respiratory distress with tachypnea, desaturation, and pleural effusion, and one (20%) because of septic shock and MOF. Two (40%) patients hospitalized in PICU also had viral coinfections (adenovirus and hMPV) detected with nasopharyngeal swabs. None had recent or ongoing VZV, RSV, Rhinovirus, or Influenza virus infections. All (100%) patients were treated with IV antibiotic therapy based on the first clinical suspect and then on the microbial sensitivity of GAS. There were no deaths among the children presented in this case series. All (100%) patients recovered and were discharged without permanent sequelae. According to the Italian vaccination program, all (100%) children were regularly vaccinated.

## 4. Discussion

From January 2012 to December 2023, we observed nine cases of iGAS infection. Four (44%) occurred before 2023 and were soft tissue infections. They had a distribution in the spring and early summer (April to June). The most significant number of GAS infections was observed in 2016. The second-highest peak was observed in 2023. Five iGAS cases were registered in the final winter–initial spring season (February–March–April) of 2023. Although the global trend in GAS infections was substantially maintained, iGAS cases significantly increased in 2023, particularly compared to the rest of the years considered in this study (5/60 [8.3%] in 2023 vs. 1/65 [1.5%] in 2016 vs. 4/371 [1.1%] in 2012–2022). 

Compared to the published reports, we noted a different seasonality. A significant rise in GAS infections was detected starting from January 2023. Notably, iGAS cases were concentrated in late winter—early spring, as reported in the Italian case series [18]. In contrast, the peak of iGAS cases in the UK, France, Ireland, Sweden, and Spain was observed in the autumn. In the Netherlands, the peak of iGAS was already observed in early 2022 [9]. Unlike in other European countries, in Italy, the wave of iGAS infections was not anticipated [17,18]. The median age of patients affected was five years, which aligns with the data from the other countries [4,8,10,11,12].

In our case series, two children had a lower respiratory tract infection with pleural effusion. Pleural empyema is the most common clinical presentation of iGAS infection in young children [4,5,6,7,10]. A multicenter study, including data from several European countries from 2012 to 2016, demonstrated that lower respiratory tract infections were risk factors for PICU admission and more severe disease courses. In addition, infections without a primary focus resulted in severe disease [16]. In our cohort, a primary focus of infection was identified in all patients. 

Viral coinfections (RSV, Influenza virus, hMPV, and Rhinovirus) have been frequently reported [4,6,8,9]. In our cohort, two patients were co-infected with hMPV and Adenovirus. This finding highlights that viral respiratory tract infections are a risk factor for more severe disease courses. No cases of previous or concurrent VZV infection were detected among our patients. 

In our cohort, *S. pyogenes* strains isolated from blood and other sterile sites were multi-sensitive, as reported in the literature, and in line with the epidemiology of antibiotic resistance in our country. 

Serum procalcitonin values were higher in children admitted to the PICU compared to those hospitalized in the inpatient department, suggesting that this marker may predict a more severe disease course at admission. Boeddha et al. noted that CRP was significantly higher in patients who required PICU admission [20]. The patient with GAS meningitis and pansinusitis had an average value of procalcitonin, and GAS was only isolated from the cerebrospinal fluid culture. The hypothesis is that the infection spread locally from the paranasal sinuses. 

At the admission, all patients showed tachycardia disproportionate to the body temperature. Vomiting was a very recurrent sign among the enrolled patients. Lymphopenia and hyponatremia were laboratory indices presented in all enrolled children, confirming their role as markers of systemic inflammatory response syndrome. These clinical and laboratory data have a relevant clinical impact because they may help clinicians recognize children who should be carefully evaluated for invasive infectious diseases early. 

All these aspects need to be taken into consideration, especially following the recent report from Vieira et al., who identified the *S. pyogenes* lineage M1_UK_ to be the dominant source of invasive infections in the recent upsurge in UK [21,22].

## 5. Conclusions

iGAS cases increased in 2023, parallel with the scarlet fever and pharyngitis cases. In the context of rising GAS infections reported internationally, we reported the clinical features of iGAS cases managed in our pediatric hospital. In a period marked by the spread of several infectious diseases, dramatically reduced during the COVID-19 pandemic, identifying the suggestive clinical features (pre-school-aged children with high fever, unexplained tachycardia, and vomiting) or predictive biomarkers (high procalcitonin levels, hyponatremia, and lymphopenia) may help distinguish children who potentially have an invasive infection and require intensive care. 

## Figures and Tables

**Figure 1 children-11-00614-f001:**
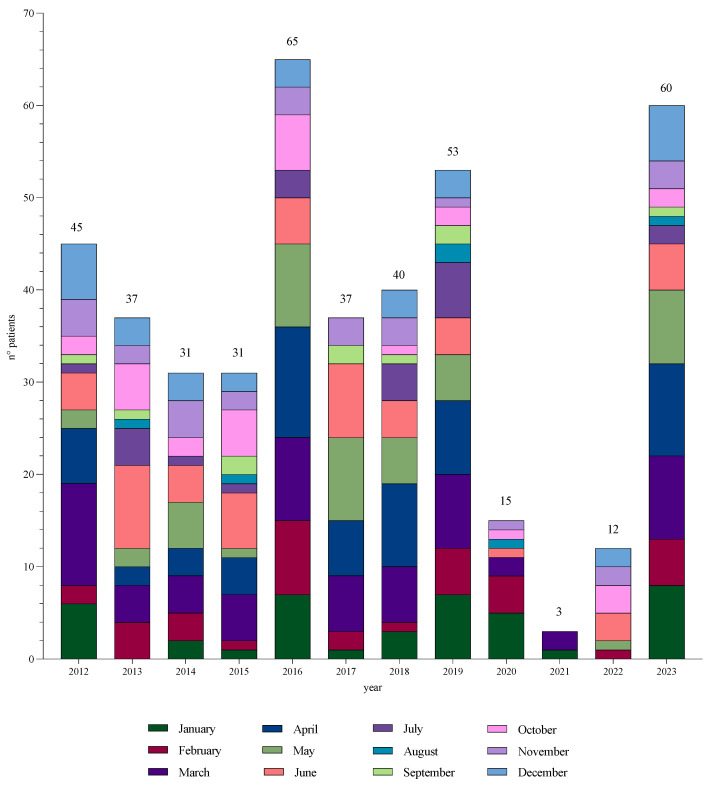
Trends of GAS infections from January 2012 to December 2023 in our pediatric clinic. Each color indicates a specific month of the year. The number of patients represents all GAS infections (including non-invasive and invasive cases) from 2012 to 2023. The total number of GAS infections corresponds to the height of the histogram and is shown at the top. The number of cases/month corresponds to the height of the month. The *Y*-axis (number of patients) is graduated every two patients. The *X*-axis shows the years from 2012 to 2023.

**Table 1 children-11-00614-t001:** Clinical features of patients with iGAS infection hospitalized from January 2012 to December 2022.

	Case 1	Case 2	Case 3	Case 4
**Year of hospitalization**	2016	2016	2019	2022
**Month of hospitalization**	April	April	April	June
**Sex**	Female	Male	Male	Male
**Age at diagnosis, years**	1	12	3	10
**Ethnicity**	Caucasian	Caucasian	Caucasian	Caucasian
**Vaccination status**	According to schedule	According to schedule	According to schedule	According to schedule
**Comorbidities**	None	None	None	Autism spectrum disorder
**Ongoing antibiotic therapy**	None	None	None	None
**Clinical manifestations**	Right ankle edema, fever, refuse to walk	Right knee swelling at the wound site, purulent discharge, fever	Left leg edema and erythema, fever	Left knee edema, fever, refuse to walk
**Infection site**	Blood	Skin	Skin	Skin
**Lab findings:** **WBC, cells/mmc** **ANC, cells/mmc** **Hb, g/dL** **PLT, cells/mmc** **CRP, mg/dL** **PCT, ng/mL**	12,550 10,800 9.5 312,000 30.8 18.4	20,440 17,000 12.2 268,000 4.6 n.a.	22,460 16,000 10.9 457,000 14 6.12	17,100 13,400 13.3 411,000 9.8 0.1
**Positive blood culture**	Yes	No	No	No
**Coinfections**	None	None	*Moraxella catarrhalis*	None
**Diagnosis**	Right ankle cellulitis in the course of GAS sepsis	Posttraumatic GAS cellulitis	Left leg GAS cellulitis	Posttraumatic GAS abscess under the knee flap
**Antibiotic therapy and duration**	Ceftriaxone for 3 days, then Teicoplanin for 10 days (based on the antibiogram) Amoxicillin-clavulanic acid for 5 days at discharge	Ceftriaxone for 5 days + Metronidazole for 5 days Amoxicillin-clavulanic acid for 10 days at discharge Metronidazole for 7 days at discharge	Ceftriaxone for 9 days + Teicoplanin for 8 days	Oxacillin for 8 days Amoxicillin-clavulanic acid for 5 days at discharge
**Complications**	None	None	None	Surgical drainage

ANC: absolute neutrophil count; CRP: C-reactive protein; Hgb: hemoglobin; PCT: procalcitonin; PLT: platelets; WBC: white blood count. n.a.: not available.

**Table 2 children-11-00614-t002:** Clinical features of patients with iGAS infection hospitalized in 2023.

	Case 5	Case 6	Case 7	Case 8	Case 9
**Month of hospitalization**	March	April	April	April	February
**Sex**	Male	Male	Male	Female	Female
**Age at diagnosis, years**	5	5	4	6	1
**Ethnicity**	Caucasian	Caucasian	Caucasian	African	Caucasian
**Vaccination status**	According to schedule	According to schedule	According to schedule	According to schedule	According to schedule
**Comorbidities**	None	Allergic rhinitis	Late preterm, esophageal, and rectal atresia, multi-cystic kidney	None	None
**Ongoing antibiotic therapy**	None	None	None	None	None
**Clinical manifestations**	Fever, ear pain, headache, vomiting, diarrhea	Respiratory distress, fever, vomiting, diarrhea, abdominal pain	Fever, vomiting, diarrhea, abdominal pain, sore throat	Fever, headache, vomit, nuchal, trunk, and limb stiffness	Respiratory distress, fever
**Vital parameters** **HR, bpm** **SpO2, %** **BT, °C** **RR, bpm** **BP, mmHg**	175 97 37.9 20 100/60	153 90 38.2 50 n.a.	144 96 37.6 20 110/55	153 96 39.2 25 n.a.	198 89 38 60 n.a.
**Infection site**	Middle ear and mastoid	Pleural fluid	Pharynx	Spinal fluid	Pleural fluid
**Positive blood culture**	Yes	Yes	Yes	No	n.a.
**WBC, cells/mmc**	9410	2150	11,150	19,620	22,610
**ANC, cells/mmc**	8660	1540	9080	18,230	17,600
**ALC, cells/mmc**	370	410	890	570	3800
**AMC, cells/mmc**	200	180	700	820	1113
**Hgb, g/dL**	12.9	12.4	12.2	10.4	12.8
**PLT, cells/mmc**	138,000	185,000	217,000	376,000	254,000
**CRP, mg/dL**	15.3	14.5	34	16.3	30.8
**PCT, ng/mL**	640	132	162	0.6	n.a.
**ALT, mU/mL**	4624	17	18	70	13
**AST, mU/mL**	10,722	52	27	21	44
**Uremia, mg/dL**	88	31	63	19	30
**Creatinine, mg/dL**	2.19	0.46	0.77	0.34	n.a.
**Capillary pH**	7.28	7.38	7.39	7.49	n.a.
**Lactate, mmol/L**	9	4.2	1.4	2.7	n.a.
**HCO3^−^, mmol/L**	14	18.5	19.2	24.2	n.a.
**Na^+^, mmol/L**	132	132	132	134	127
**Coinfections**	None	Adenovirus, Metapneumovirus	Adenovirus	n.a.	None
**Diagnosis**	Severe GAS sepsis with septic shock and MOF in patients with mastoiditis complicated by venous sinus thrombosis	Left pneumonia with bilateral pleural effusion in GAS infection	Pharyngotonsillitis and GAS sepsis	GAS meningitis	GAS pleural empyema and respiratory failure
**Antibiotic therapy and duration**	Ceftriaxone for 9 days Clindamycin for 3 weeks	Ceftriaxone for 17 days Linezolid for 13 days Metronidazole for 13 days	Piperacillin-tazobactam for 10 days, Amoxicillin-clavulanic acid for 5 days at discharge	Ceftriaxone for 10 days	Ampicillin-sulbactam for 7 days Ceftriaxone and vancomycin for 8 days Cefixime for 8 days at discharge
**Complications**	Sigmoid sinus and jugular vein thrombosis, septic shock with MOF	Pleural effusion, need of ventilative support (CPAP, HHHFNC)	None	None	Pleural empyema, the need for invasive ventilative support
**ICU admission**	Yes	Yes	No	No	Yes

BP: blood pressure; BT: body temperature; CPAP: continuous positive airway pressure; GAS: group A streptococcus; HHHFNC: heated humidified high-flow nasal cannula; HR: heart rate; ICU: intensive care unit; MOF: multiorgan failure; RR: respiratory rate. WBC: white blood count; ANC: absolute neutrophil count; ALC: absolute lymphocyte count; AMC: absolute monocyte count; Hgb: hemoglobin; PLT: platelets; CRP: C-reactive protein; PCT: procalcitonin; ALT: alanine aminotransferase; AST: aspartate aminotransferase. n.a.: not available.

## Data Availability

The data presented in this study are available on request from the corresponding author (riccardo.castagnoli@unipv.it) due to privacy.

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
