# Peer review of "Invasive Streptococcal Infection in Children: An Italian Case Series"

_children, 2024, doi:10.3390/children11060614_

Round 1

Reviewer 1 Report

Comments and Suggestions for Authors

The present article is definitely useful for timely recognising and treating GAS cases, and it deserves good English.

Comments on the Quality of English Language

Τhe language of the paper can be somehow improved.

Author Response

“The present article is definitely useful for timely recognising and treating GAS cases, and it deserves good English. Τhe language of the paper can be somehow improved.”

We thank the reviewer for the positive comments. We provided extensive language editing. 

Reviewer 2 Report

Comments and Suggestions for Authors

The paper is in accordance with internal reports on the increase in iGAS cases. It is important to share data about these trends as specific disease intervention is needed.

The paper is well-written, with good support to all the findings. My main concern is that the authors do not present any number, code or protocol name for the Study approval. It is not enough to state that the study was approved. Internation regulations on ethics in human study and experimentation require the authors to provide enough information that can be tracked about the approval.

Author Response

“The paper is in accordance with internal reports on the increase in iGAS cases. It is important to share data about these trends as specific disease intervention is needed.

The paper is well-written, with good support to all the findings. My main concern is that the authors do not present any number, code or protocol name for the Study approval. It is not enough to state that the study was approved. Internation regulations on ethics in human study and experimentation require the authors to provide enough information that can be tracked about the approval.”

We thank the reviewer for the positive comments. We added more information about the study protocol. Please see lines 135-139.

Reviewer 3 Report

Comments and Suggestions for Authors

The text is interesting; however, it poorly explains the increase in the number of cases in the center. As the reviewer understood, there was only one case. So how do we know that there has been an increase in the number of cases? The figures are not very clear. The recording of pathogens needs correction. The authors should clarify when to use uppercase and italics.

Comments on the Quality of English Language

The text is interesting; however, it poorly explains the increase in the number of cases in the center. As the reviewer understood, there was only one case. So how do we know that there has been an increase in the number of cases? The figures are not very clear. The recording of pathogens needs correction. The authors should clarify when to use uppercase and italics.

Author Response

The text is interesting; however, it poorly explains the increase in the number of cases in the center. As the reviewer understood, there was only one case. So how do we know that there has been an increase in the number of cases? The figures are not very clear. The recording of pathogens needs correction. The authors should clarify when to use uppercase and italics.

We thank the reviewer for the comments. Our manuscript is mostly focused on invasive streptococcal infections that were compared to the overall number of GAS infections documented in our Pediatric Clinic from 2012 to 2023. We reported that invasive iGAS infections were prevalent in 2023, compared to the rest of the years considered in this study (5/60 [8.3%] cases in 2023 vs. 1/65 [1.5%] in 2016 vs. 4/371 [1.1%] in 2012-2022). We described Figure 1 explaining how to interpret the data and histograms. We corrected the names of pathogens and the use of uppercase and italics. Please see the revised manuscript.